# A High-Efficiency Modular Multiplication Digital Signal Processing for Lattice-Based Post-Quantum Cryptography

**Trong-Hung Nguyen** *[ID], **Cong-Kha Pham** [ID] and **Trong-Thuc Hoang** [ID]

Department of Computer and Network Engineering, University of Electro-Communications (UEC), 1-5-1 Chofugaoka, Tokyo 182-8585, Japan; phamck@uec.ac.jp (C.-K. P.); hoangtt@uec.ac.jp (T.-T. H.)
* Correspondence: tronghung@vlsilab.ee.uec.ac.jp

**Abstract:** The Number Theoretic Transform (NTT) has been widely used to speed up polynomial multiplication in lattice-based post-quantum algorithms. All NTT operands use modular arithmetic, especially modular multiplication, which significantly influences NTT hardware implementation efficiency. Until now, most hardware implementations used Digital Signal Processing (DSP) to multiply two integers and optimally perform modulo computations from the multiplication product. This paper presents a customized Lattice-DSP (L-DSP) for modular multiplication based on the Karatsuba algorithm, Vedic multiplier, and modular reduction methods. The proposed L-DSP performs both integer multiplication and modular reduction simultaneously for lattice-based cryptography. As a result, the speed and area efficiency of the L-DSPs are 283 MHz for 77 SLICEs, 272 MHz for 87 SLICEs, and 256 MHz for 101 SLICEs with the parameters $q$ of 3329, 7681, and 12,289, respectively. In addition, the $N^{-1}$ multiplier in the Inverse-NTT (INTT) calculation is also eliminated, reducing the size of the Butterfly Unit (BU) in CRYSTAL-Kyber to about 104 SLICEs, equivalent to a conventional multiplication in the other studies. Based on the proposed DSP, a Point-Wise Matrix Multiplication (PWMM) architecture for CRYSTAL-Kyber is designed on a hardware footprint equivalent to 386 SLICEs. Furthermore, this research is the first DSP designed for lattice-based Post-quantum Cryptography (PQC) modular multiplication.

**Keywords:** post-quantum cryptography (PQC); lattice-based cryptography (LBC); CRYSTAL-Kyber; FALCON; number theoretic transform (NTT); point-wise-matrix-multiplication (PWMM)

## 1. Introduction

In 2016, the National Institute of Standards and Technology (NIST) initiated the PQC standardization process. This project aims to develop, deploy, and standardize new post-quantum cryptosystems before any large-scale quantum computers come into being. In July 2022, NIST announced the results of the third round with four candidates to be standardized for Public Key Encryption (PKE) and Digital Signature Algorithm (DSA) [1]. Most are Lattice-Based Cryptographic (LBC) algorithms, CRYSTAL-Kyber [2], and CRYSTAL-Dilithium/FALCON [3,4], respectively.

Lattice-based cryptographic constructions are primarily based on solving the Learning-With-Error (LWE) and its variants problem (CRYSTAL-Kyber, Dilithium) or NTRU lattices (FALCON). Implementing the LBC cryptosystem requires performing polynomial multiplication, the most hardware-intensive operation. There are two ways to do polynomial multiplication: the Schoolbook polynomial multiplication and the multiplication based on NTT. Schoolbook multiplication is inefficient for polynomial multiplication because it has a $O(N^2)$ complexity. NTT is the special Discrete Fourier Transform case over a finite field. NTT-based multiplication enhances polynomial multiplication, reducing $O(N^2)$ complexity to quasi-linear complexity $O(N \cdot logN)$. In order to improve the efficiency of the LBC cryptosystem, NTT optimization is necessary. Furthermore, all NTT operations are modulo operations on prime $q$, including modular addition, subtraction, and multiplication. Performing modular multiplication is a highly intricate task that demands

significant hardware resources. Thus, improving modular multiplication can enhance the performance of NTT/INTT and the entire cryptosystem. The optimization problem of modulo computations for multiplication has garnered significant attention in hardware implementations of LBC post-quantum cryptography research.

The accelerator proposals for NTT/INTT are mainly focused on optimizing modulo calculations from the product of multiplying two integers. Montgomery [5] and Barrett [6] are two commonly used constant-time modular reduction algorithms. The Montgomery method has received less attention as it needs to be done in the "Montgomery domain". On the other hand, the Barret method is more efficient and is used more frequently in LBC cryptosystems [7]. Barrett reduction utilizes pre-computed values to approximate the division by the modulus. The modular multiplication based on the Barret method requires three multiplications, one for the two input coefficients and two for the constants. A variation of the Barret algorithm in [8], called Shift-Add-Multiply-Subtract-Subtract (SAMS2), replaces constant multipliers with simple bit shifts, additions, and subtractions, which are less expensive than multiplication and division operands. Studies [9–11] have applied the SAMS2 method for parameters $q$ = 7681 and 12,289.

In the study [12], Plantard introduced a novel constant-time modular reduction algorithm. Like the Montgomery and Barrett algorithms, Plantard multiplication utilizes pre-computed values and requires three multiplications for modular multiplication. But, when performing NTT/INTT with pre-computed twiddle factors, the number of Plantard multiplications can be reduced by one. In another study [13], J. Huang et al. enhanced the Plantard method to accommodate signed integers as input and narrowed the range of the modulus $q$ to $\left(-\frac{q}{2}; \frac{q}{2}\right)$.

Another effective method for reducing modulus in the LBC system is to utilize the characteristic property of prime number $q$ [14,15]. This approach enables modular calculations through lightweight operations, including bit-wise, addition, and subtraction. An alternative and more straightforward method for high-order bits modulus $q$ is using the look-up tables, as demonstrated in [16].

In a different research study [17], Longa et al. proposed a method called K-RED, which utilizes a special format of NTT-applicable primes, $q = k \cdot 2^m + 1$. This method primarily includes multiplying by a small coefficient ($k$) and subtracting, resulting in significantly lower computational costs than other methods. The product input $c = a \cdot b$ is reduced to the signed integer $r \equiv k \cdot c \pmod q$. In study [18], Bisheh-Niasar et al. based on the K-RED method and introduced the $K^2$-RED method by applying K-RED twice for CRYSTAL-Kyber. Furthermore, the cumulative coefficients, $k/k^2$, can be eliminated by merging $k^{-1}/k^{-2}$ into the twiddle factor $\omega/\omega^{-1}$ in NTT/INTT processes. In particular, in study [19], Li et al. used the $-k \cdot 2^m \equiv 1 \pmod q$ property of the modulo q calculation, which helps to apply the K-RED method to the Point-Wise-Multiplication (PWM) process. Additionally, the multiplication by a factor of $N^{-1}$ has been removed in the INTT process.

Nevertheless, as far as we know, studies presently focus on optimizing from the input multiplication. This study aims to design a DSP for modular multiplication by developing a multiplication unit for two integer inputs and utilizing advanced modular reduction techniques. In particular, the Karatsuba algorithm is used to subdivide the size of multiplication by half. Partial multiplications are performed using the Vertical and Crosswise algorithm, which speeds up computation. The results of the multiplications are reduced to bit-widths following the K-RED method's condition using the pre-computed look-up table. Finally, the K-RED method is applied to calculate the result of the modulo operation, $a \cdot b \ (mod \ q)$.

The major contributions of this paper are as follows:

- Proposes the first specialized DSP that performs modular multiplication for the CRYSTAL-Kyber PQC algorithm, called Kyber-DSP (K-DSP).
  The K-DSP performs the multiplication of two input integers and modular reduction for the prime $q$ = 3329. The architecture reaches a high frequency of 283 MHz, and

the area is only 77 SLICEs, equivalent to 77% of a typical DSP. This result completely outperforms traditional methods of modular multiplication that rely on DSP.

- The proposed Lattice-DSP (L-DSP) configuration optimizes the BU in NTT/INTT. In addition to saving on hardware resources, using the proposed L-DSP also eliminates the $N^{-1}$ multiplication in the INTT process. As a result, the BU architecture requires minimal hardware resources. Choosing the architecture for NTT accelerators based on Decimation-In-Time (DIT), Decimation-In-Frequency (DIF), or both has become more flexible and easier. In CRYSTAL-Kyber, the BU architecture reaches a high frequency of 283 MHz while occupying an area equivalent to one DSP.

- Designs a K-DSP-based PWMM architecture designed for CRYSTAL-Kyber. PWM calculation in CRYSTAL-Kyber is more complicated than other LBC algorithms, requiring at least four multiplications for two PWM results. This study introduces a specific PWM structure for CRYSTAL-Kyber that uses K-DSP. Furthermore, the cumulative computation of matrix multiplication is combined with PWM while maintaining the same hardware cost for all three Kyber security levels (1, 3, and 5). The architecture that implements PWMM on the NTT domain includes PWM and Point-Wise Addition (PWA). The proposed PWMM operating frequency reaches 275 MHz with a hardware area of 386 SLICEs, equivalent to closely 4 DSPs.

- Extended with L-DSP design for prime numbers $q = 7681$ and 12,289. The proposed DSP design method is ideal for NTT-friendly algorithms with a prime factor $q = k \cdot 2^m + 1$. By applying this design to the case where $q = 7681$ and 12,889, it has been proven that the method still allows for a high operating frequency of 272 MHz and 256 MHz while using 87 SLICEs and 101 SLICEs of hardware resources.

The remainder of the paper is organized as follows. Section 2 introduces the theoretical background of LBC, specifically the CRYSTAL-Kyber algorithm, and describes the NTT-based polynomial multiplication. Section 3 discusses in more detail the existing implementation studies for modular reduction. Section 4 presents the implementation of a DSP design for modular multiplication and builds upon the BU and PWMM architectures. Section 5 compares the performance of the proposed DSP and the designs built on it with the state-of-the-art reference implementations of Field-Programmable Gate Arrays (FPGAs). Finally, in Section 6, the conclusion of the paper is presented.

## 2. The Background

In this section, the CRYSTAL-Kyber algorithm is selected as the research basis for the improvement proposals applicable to other LBC algorithms. From a hardware implementation perspective, CRYSTAL-Kyber is designed to do all the polynomial multiplication using NTT. Also, the parameter set of CRYSTAL-Kyber is the smallest with $(N, q) = (256, 3329)$, and PWM operation requires more difficult hardware complexity.

Unlike other LBCs, FALCON is very complex in hardware implementation due to having to perform polynomial multiplication on complex and integer number domains [20]. Fortunately, verification of FALCON is simple and high-performing. This crucial advantage led NIST to select FALCON for standardization with Dilithium after the third-round finalist. Thus, Kyber's improvements to polynomial multiplication over the integer domain can be applied to enhance FALCON's verification performance.

### 2.1. CRYSTAL-Kyber

CRYSTAL-Kyber is built based on the hardness of the Module-LWE problem. Kyber employs a two-stage construction for achieving an INDistinguishability under a Chosen-Ciphertext Attack (IND-CCA) secure Key-Encapsulation Mechanism (KEM). First, an IND-Chosen Plaintext Attack (CPA)-secure public-key encryption scheme is built called Kyber.CPA. Then, the Fujisaki-Okamoto transform is applied to build the CCA-secure KEM.

Kyber operates on $\mathbf{R}_q = \mathbb{Z}_q[X]/(X^N + 1)$ with module rank $k$ = 2, 3, and 4, providing security levels 1, 3, and 5 for Kyber512, Kyber768, and Kyber1024, respectively. In the first round of NIST [21], the prime $q$ was chosen as 7681 and satisfying $q \equiv 1 (mod\ 2N)$,

where $N = 256$ is the degree of the modular polynomial. In the second round of the NIST competition [22], the Kyber team reduced the $q$ value from 7681 to 3329 as research [23] showed that this value of $q$ also supports very fast NTT-based polynomial multiplication. Simultaneously, $q = 3329$ also requires a smaller noise range while maintaining the same level of security as Kyber(v01). Table 1 lists the parameters of Kyber(v03), where $\eta_1$ and $\eta_2$ are the parameters of the central binomial distribution.

**Table 1.** Parameter sets for CRYSTAL-Kyber (v03).

|  | **N** | **q** | **k** | **$\eta_1$** | **$\eta_2$** |
|---|---|---|---|---|---|
| Kyber512 | 256 | 3329 | 2 | 3 | 2 |
| Kyber768 | 256 | 3329 | 3 | 2 | 2 |
| Kyber1024 | 256 | 3329 | 4 | 2 | 2 |

*2.2. NTT-Based Polynomial Multiplication*

The NTT-based multiplication is commonly used for LBCs. It involves transforming polynomials from coefficient representation to the NTT domain, performing pointwise operations, and then returning the result to the integer domain using the INTT transformation. Using NTT, the multiplication of two integer polynomials with $N$ terms can be performed with a low computational cost of $O(N \cdot logN)$. Especially, the prime $q$ must be chosen with condition $q \equiv 1 (mod\ 2N)$ to avoid zero-padding when performing NTT-based multiplication, which is also known as Negative-Wrap Convolution (NWC) [24]. For the prime $q = 3329$ of Kyber(v02/03), the field $\mathbb{Z}_q$ contains primitive $N$-th roots of unity but not primitive $2N$-th roots. Thus, the NTT of polynomial $f \in \mathbf{R}_q$ is a vector of 128 degree-1 polynomials with two coefficients each and is defined as,

$$NTT(f) = \hat{f} = (\hat{f}_0 + \hat{f}_1 X, \hat{f}_2 + \hat{f}_3 X, \ldots, \hat{f}_{254} + \hat{f}_{255} X), \tag{1}$$

where coefficients $\hat{f}_i$ are defined as,

$$\hat{f}_{2i} = \sum_{j=0}^{127} f_{2j} \zeta^{(2br_7(i)+1)j} \tag{2}$$

$$\hat{f}_{2i+1} = \sum_{j=0}^{127} f_{2j+1} \zeta^{(2br_7(i)+1)j} \tag{3}$$

with $\zeta$ is the 256-th root of unity in $\mathbb{Z}_q$ and $i = 0, \ldots, 127$ is a 7-bit unsigned integer represented in binary form as $i = [i_6, i_5, i_4, i_3, i_2, i_1, i_0]$. $br_7(i)$ is the bit reversal of $i$, where $br_7(i) = [i_0, i_1, i_2, i_3, i_4, i_5, i_6]$. As a result, a 256-term polynomial in Kyber is divided into two 128-term polynomials by parity, and NTT is applied to each one. After NTT, the PWM of the two components $f$ and $g$ of $\mathbf{R}_q$, denoted by $\hat{f} \circ \hat{g}$, is performed by conducting 128 multiplications of linear polynomials modulo $X^2 - \zeta^{2br(i)+1}$. Specifically,

$$h = f \cdot g = INTT(NTT(f) \circ NTT(g)), \tag{4}$$

where $\hat{h} = \hat{f} \circ \hat{g}$ is defined as,

$$\hat{h} = \hat{h}_{2i} + \hat{h}_{2i+1} X = (\hat{f}_{2i} + \hat{f}_{2i+1} X)(\hat{g}_{2i} + \hat{g}_{2i+1} X)\ mod(X^2 - \zeta^{2br(i)+1}) \tag{5}$$

Finally, the point-wise multiplications in Kyber are defined as,

$$\hat{h}_{2i} = \hat{f}_{2i} \hat{g}_{2i} + \hat{f}_{2i+1} \hat{g}_{2i+1} \cdot \zeta^{2br(i)+1} \tag{6}$$

$$\hat{h}_{2i+1} = \hat{f}_{2i} \hat{g}_{2i+1} + \hat{f}_{2i+1} \hat{g}_{2i} \tag{7}$$

### 3. Related Works

LBC operations are performed on the ring $\mathbf{R}_q = \mathbb{Z}_q[X]/(X^N + 1)$, where $q$ is a prime number and $N$ is a power-of-two. Modular multiplication is the most time-consuming operand in NTT and can be expressed as follows:

$$a \times b = c \equiv x \bmod q \quad (0 \leq a, b, x < q;\ 0 \leq c < q^2) \tag{8}$$

Several classical algorithms are available to enhance the efficiency of modular reduction, such as Montgomery reduction and Barret reduction. The Montgomery method is infrequently used due to the resource consumption of conversions into and out of the "Montgomery domain" [25,26]. In contrast, the Barret method is widely adopted and has many improved variants. The basic idea behind Barret's algorithm is to pre-compute the inverse of modulus $q$ and use simple bit shifting and multiplication instead of costly division. Algorithm 1 utilizes Barrett reduction to compute the product of two integers modulo $q$.

---

**Algorithm 1** Modular Multiplication by Barret Reduction [7]

---

**Input:** $a, b, q \in \mathbb{Z}$
**Output:** $a \times b \ (mod\ q)$
    **Pre-computation**
1:  $k = \lceil log_2\ q \rceil$;
2:  $r = 2^k$;
3:  $\mu = \left\lfloor \frac{r^2}{q} \right\rfloor$;
    **Multiplication**
4:  $z = a \times b$;
    **Barret reduction**
5:  $m_1 = \left\lfloor \frac{z}{r} \right\rfloor$;
6:  $m_2 = m_1 \times \mu$;
7:  $m_3 = \left\lfloor \frac{m_2}{r} \right\rfloor$;
8:  $t = z - m_3 \times q$;
9:  **if** $t \geq q$ **then**
10:      **return** $t - q$
11: **else**
12:      **return** $t$
13: **end if**

---

In LBC algorithms, the $q$ value is fixed, allowing pre-computation of $k$, $r$, and $\mu$. Barrett-based modular multiplication commonly employs DSPs for multiplying input coefficients and the constant $\mu$, while multiplying by $q$ is efficiently achieved using bitwise shifts and additions. In study [27], Dang et al. applied a variation of Barret reduction in [28] to select parameter values $(\alpha, \beta)$ and design a single-constant-multiplier for multiplying by the constant $\mu$. LBC schemes based on NTT implementation using signed integers can eliminate modular addition at each butterfly unit [25,29,30]. The optimized Barrett reduction algorithm for signed integer inputs has been further enhanced by study [31] to narrow its output range to $\left(-\frac{q}{2}, \frac{q}{2}\right)$. This improvement effectively limits the growth of coefficients after each butterfly unit, resulting in better performance. The SAMS2 method simplifies multiplication by bit shifting, addition, and subtraction [8–11]. This significantly reduces the hardware architecture but increases latency due to multiple subtractions. A look-up table can be used to speed up, but it is inefficient in terms of area.

Huang et al. enhanced the Plantard algorithm for a larger range of inputs and a smaller range of outputs [13]. The improved Plantard method saves one multiplication compared to the latest Montgomery and Barrett methods. However, the drawback of this method is that it still requires three multiplications when calculating the PWM and necessitates doubling the width of the pre-computed intermediate twiddle factors.

Several methods have been proposed for modular reduction to optimize the area and speed of NTT accelerator with specific $q$ parameter. Study [14] utilized the form of $q = 2^{l_1} \pm 2^{l_2} \pm \cdots \pm 1$ to replace multiplications in Barret reduction with bit shifts, addition, and subtraction operations. Aikata et al. implemented this technique for the Kyber $q = 3329$ case [32]. Some recent studies involve calculating the modulus $q$ of higher-order bits in the product of multiplying two 12-bit integers, $c[23 : 0] = a \times b$. In studies [15,33], the property $2^{12} \equiv 2^9 + 2^8 - 1 \, (mod \, 3329)$ is used to gradually reduce the higher-order bits to an arithmetic combination of the smaller bit arrays. Similarly, in study [34], the bit width of the multiplication product is reduced from 24 to 15, and then apply Barret algorithm. This method is useful in reducing the multiplication size during Barret reduction. Additionally, a different format of $q = \delta \cdot 2^e + 1$, is used to propose a modulus reduction algorithm for Kyber [35]. This algorithm divides the product $c$ into two corresponding parts, $c = c_1 \cdot 2^e + c_0$, and replaces the large modulus $q$ with the smaller modulus $\delta$. A simpler and more efficient alternative is introduced in [16]. Zhang et al. used the pre-computed look-up table to store the calculations of $c[23 : 20] \cdot 2^{20} (mod \, 3329)$, $c[19 : 16] \cdot 2^{16} (mod \, 3329)$, and $c[15 : 12] \cdot 2^{12} (mod \, 3329)$. The higher-order bits are used as the input address of the look-up tables, and the outputs are the corresponding modular operations. Finally, the modulo operation of the product $c[23 : 0]$ is calculated by adding the four numbers on the ring $\mathbf{R}_{3329}$.

Another new modular reduction approach, K-RED, is proposed in study [17]. The K-RED method utilizes the characteristics of Proth numbers represented as $q = k \cdot 2^m + 1$ where $k$ is a small number, $m$ is a natural number. This approach presents two functions: K-RED and KRED-2X, which take any integer $c$ and return an integer $d$ such that $d \equiv k \cdot c \, mod \, q$ and $d \equiv k^2 \cdot c \, mod \, q$, respectively. The K-RED method performs one multiplication with a constant factor of $k$ and one subtraction, as described in Algorithm 2. Multiplying by $k$ is achieved with bit shifting and addition, significantly reducing computational costs compared to other methods. However, to correct the reduction results, the output of K-RED must be multiplied by the factor $k^{-1}/k^{-2}$. Bisheh-Niasar et al. in [18] followed this scheme and proposed $K^2$-RED, by applying K-RED twice in CRYSTAL-Kyber with constant $k = 13$ and $m = 8$. The multiplication by $k^{-1}/k^{-2}$ can be combined with the pre-computed twiddle factor $\omega$ in NTT/INTT for faster computation with fewer hardware resources. In study [36], Ni et al. segmented the product into two parts. The look-up table method is employed for the bits exceeding 20, while the remaining portion underwent the K-RED technique. This method is simple but requires one more adder. The $K^2$-RED is extended to $K^l$-RED for different NTT parameters, where $l = \lceil t/m \rceil$ is the number of loops, and $t$ is the bit-length of input coefficients [37]. However, K-RED is not appropriate for PWM calculations with random multipliers. Fortunately, this drawback can be resolved by using the property of $-k \cdot 2^m \equiv 1 \, (mod \, q)$, as demonstrated in study [19]. Li et al. applied the K-RED method with modifications in input value and output calculations. In particular, the input product $c$ is multiplied by two, and the subtraction in K-RED is changed sign. Therefore, the output is calculated as,

$$r \equiv 2c \; (mod \, q) \equiv (-k \cdot N) \cdot \left(\frac{N}{2}\right)^{-1} (mod \, q) \tag{9}$$

with $(k, N) = (13, 256)$ and $-13 \cdot 256 \equiv 1 \; (mod \, 3329)$ for the case of Kyber, the Equation (9) is equivalent to $r \equiv 128^{-1} \cdot c \; (mod \, 3329)$. During the INTT process, multiplying by $128^{-1}$ is considered post-processing. As a result, K-RED can be applied to PWM processes, eliminating the need for post-processing in Kyber by using this method.

---

**Algorithm 2** K-RED Modular Reduction Algorithm [17]

---

**Input:** $c$, parameter: $m,k$.
**Output:** $d \equiv k \cdot c \, (mod \, q)$
　1: $d_0 = c \, (mod \, (2^m))$;
　2: $d_1 = c/2^m$;
　3: Return $(kd_0 - d_1)$

---

## 4. Proposed Hardware Design

The modular reduction of LBCs typically begins with the DSP-based multiplication product, but small multiplication widths can result in suboptimal area usage. As a result, in this section, we present a K-DSP architecture as the basis for implementing BUs in NTT/INTT processes. Further, we propose a PWMM unit for Kyber using the K-DSP and extend the L-DSP design method to other LBC cases with $q = 7681$ and 12,289.

### 4.1. K-DSP

The K-DSP performs modular multiplication of two 12-bit coefficients, $a[11:0]$ and $b[11:0]$, to produce a 12-bit result $c[11:0]$ on the ring $\mathbf{R}_{3329}$. The proposed architecture of K-DSP is shown in Figure 1 with three computational stages. In the initial stage, we employ the Karatsuba algorithm to partition a 12-bit multiplication operation into three discrete components: two 6-bit multiplications and one 7-bit multiplication, respectively $a_H \cdot b_H$, $a_L \cdot b_L$, and $(a_H + a_L) \cdot (b_H + b_L)$. Subsequently, the summation of these partial products is calculated using the look-up table method. As a result of this procedure, the bit-width of the product is reduced from 24 bits to 20 bits. Finally, the K-RED method is utilized to ascertain the modulus $q$ of the 20-bit product.

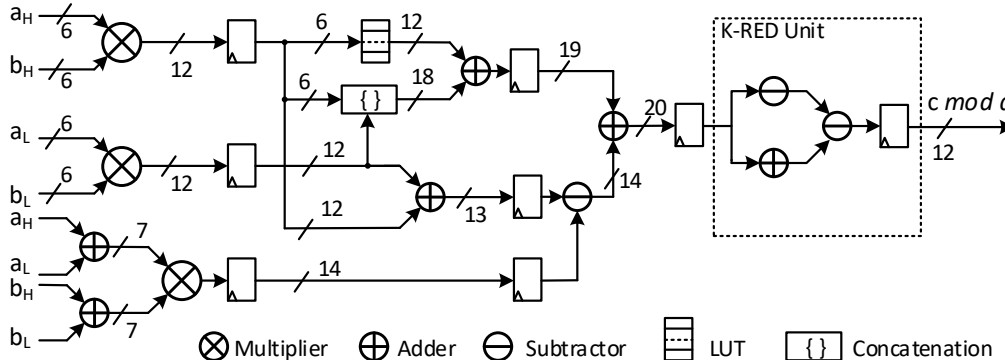

**Figure 1.** The proposed architecture of K-DSP.

The choice of multiplier design significantly impacts the speed and area of the proposed DSP. This study selects the Vedic multiplier based on the Vertical and Crosswise technique for designing 6-bit and 7-bit multipliers due to its shorter critical path than the conventional array multipliers [38]. Vedic multipliers are performed in parallel, and the partial products are added together by two or three levels of the adder [39]. Figure 2 shows the 3-bit multiplier architecture of two numbers $a[a_2, a_1, a_0]$ and $b[b_2, b_1, b_0]$. The architecture comprises nine AND gates, three full adders, and three half adders.

Figure 3 depicts the proposed 6-bit multiplier architecture based on four 3-bit multipliers and three 6-bit adder units. The adder unit used is the carry save adder to perform the additions in parallel, improving speed efficiency [40]. A 4-bit Vedic multiplier is also designed. In the K-DSP architecture, the 3-bit and 4-bit Vedic multipliers are used as the base multipliers for building up the 6-bit and 7-bit multipliers.

According to the Karatsuba algorithm, multiplication is done in three steps: $a_H \cdot b_H$, $a_L \cdot b_L$, and $(a_H + a_L) \cdot (b_H + b_L)$, then subtract $a_H \cdot b_H$ and $a_L \cdot b_L$ from the result of $(a_H + a_L) \cdot (b_H + b_L)$ to get the product. A look-up table is designed to compute the operation $[23:18] \cdot 2^{18} \bmod 3329$, with the input address being the $[23:18]$ bits of the product $a_H \cdot b_H$. The final product is then reduced modular to a bit width of 20 bits.

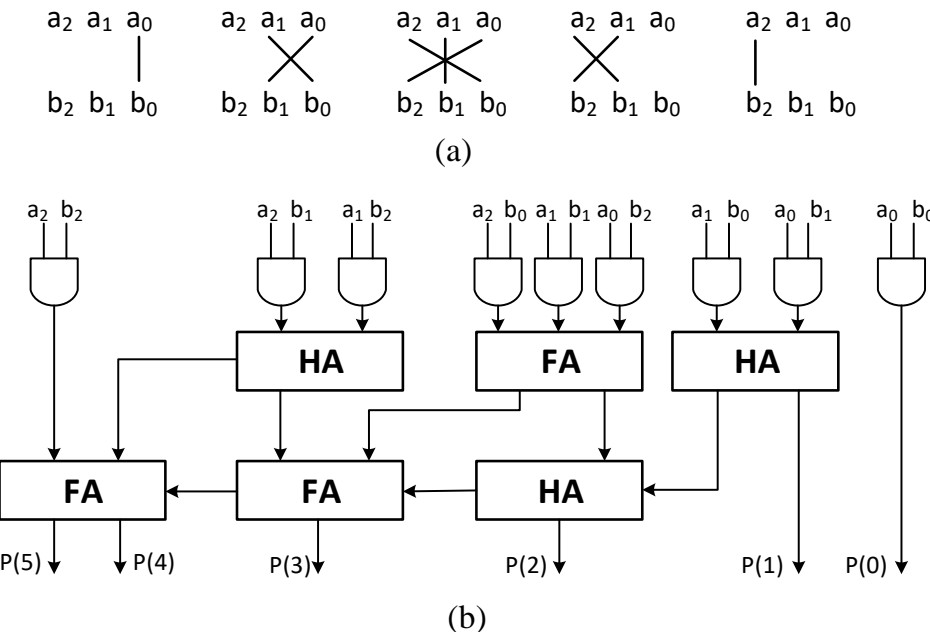

**Figure 2.** (**a**) 3-bit multiplication by Vertically and Crosswise technique. (**b**) Architecture block diagram for 3-bit Vedic multiplier.

The prime parameter in Kyber is $q = 13 \cdot 2^8 + 1$, where the factor values are $k = 13$ and $m = 8$. The elimination of accumulation is handled differently by implementing K-DSP in each BU or PWM unit. Consequently, the K-RED-based modular reduction part can be regarded as a specialized sub-module, which will be further discussed in the following sub-sections.

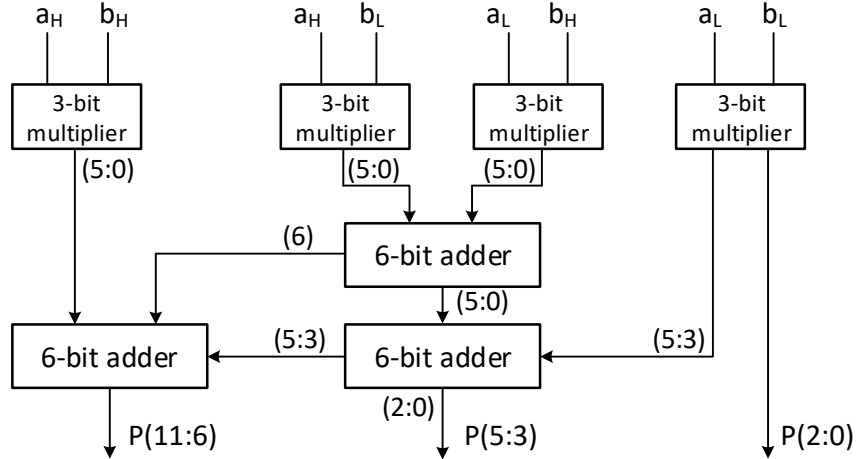

**Figure 3.** The architecture of 6-bit Vedic multiplier.

### 4.2. Butterfly Unit

The NWC technique is utilized in NTT/INTT to avoid doubling the size of the multiplication polynomial. To compute $\mathbf{c} = \mathbf{a} \times \mathbf{b}$ in the ring $\mathbf{R}_q$ with NWC, polynomials $\mathbf{a}$ and $\mathbf{b}$ must be scaled by a factor $\varphi$ before applying NTT (refered to as pre-processing). Subsequently, polynomial product $\mathbf{c}$ is scaled by a factor $N^{-1} \cdot \varphi^{-1}$ after INTT (referred to as post-processing), and $\varphi$ is the *2N*-th primitive root of the unity. Two methods for calculating the NTT are DIT and DIF, corresponding to the Cooley-Tukey (CT) [41], and Gentleman-Sande (GS) butterfly configurations [42]. Given a pair of coefficients $(a, b)$ and twiddle factor $\omega$, the CT and GS butterfly calculations produce the results of $(a + b \cdot \omega, a - b \cdot \omega)$ and $(a + b, (a - b) \cdot \omega)$, as depicted in Figure 4a,b.

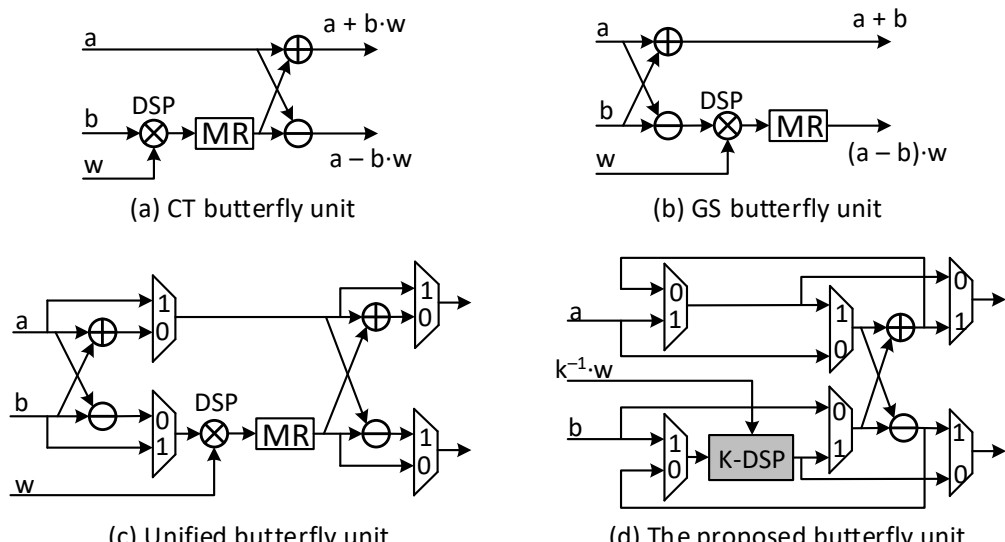

**Figure 4.** The configurations of butterfly unit.

In studies [43,44], the $\varphi/\varphi^{-1}$ multipliers can be merged with NTT and INTT processes. These methods involve separate butterfly operations: CT for NTT with $\varphi$ and GS for INTT with $\varphi^{-1}$. The unified-BU architecture has been proposed for the simultaneous computation of GS and CT in the study [14], as shown in Figure 4c. In the GS calculation, the factor $N^{-1}$ can be pre-computed with the twiddle factor $\omega$ for the $(a-b) \cdot \omega$ operation. With the operation $(a+b)$, in study [45], Zhang et al. proposed replacing the multiplication by $N^{-1}$ after INTT with the multiplication by $2^{-1}$ in each butterfly operand, respectively $(a+b)/2$. The hardware architecture for multiplier $2^{-1}$ in $\mathbf{R}_q$ is achieved simply by implementing $a/2 = (a \gg 1) + a[0] \cdot \frac{q+1}{2}$.

Butterfly operations are all modular arithmetic. The specific configuration choice among CT, GS, or both depends on the design of the NTT accelerator. The iterative NTT architecture uses unified BU for butterfly operations in all stages, such as NTT, INTT, and PWM with the Kyber case. The latency of iterative NTT increases with the number of butterfly cores and becomes more complex when handling high-order polynomials. On the other hand, the NTT pipeline architecture allows for flexibility in selecting BU configurations and butterfly cores proportional to the number of NTT stages [11,46,47]. In all cases, modular multiplication is consistently the most hardware-intensive operation and represents the critical delay path.

In this study, a BU tailored for Kyber is implemented. This architecture employs K-DSP and comprises one modular addition, one subtraction, and one multiplication. The resulting output is controlled by mux units, allowing the BU to operate in either CT or GS mode, as depicted in Figure 4d. The classical K-RED method is applied for the modular reduction part of K-DSP. The accumulated $k = 13$ is removed when the inverse $k^{-1}$ is merged into the twiddle factor $\omega$. Additionally, the multiplication by $N^{-1}$ for the GS calculation is eliminated, as performed at the PWM stage (further details are provided in the subsequent section). BUs utilize K-DSP units, helping reduce the size and improving the efficiency of parallel or pipeline architectures when multiple BUs are used.

### 4.3. Point-Wise Matrix Multiplication Unit

Calculating the ciphertext $\mathbf{u} = INTT(\hat{\mathbf{A}}^T \circ \hat{\mathbf{r}}) + \mathbf{e_1}$ is the most intricate operation in Kyber, with $\mathbf{u}$, $\mathbf{r}$, and $\mathbf{e_1}$ are vector polynomials, and $\mathbf{A}$ is matrix polynomial. The specific mathematical expression for the PWMM is shown in Equation (10) for case module rank $k = 2$.

$$\hat{\mathbf{A}}^T \circ \hat{\mathbf{r}} = \begin{bmatrix} \hat{a}_{00} & \hat{a}_{01} \\ \hat{a}_{10} & \hat{a}_{11} \end{bmatrix}^T \circ \begin{bmatrix} \hat{r}_0 \\ \hat{r}_1 \end{bmatrix} = \begin{bmatrix} \hat{a}_{00} \circ \hat{r}_0 + \hat{a}_{10} \circ \hat{r}_1 \\ \hat{a}_{01} \circ \hat{r}_0 + \hat{a}_{11} \circ \hat{r}_1 \end{bmatrix} \quad (10)$$

The PWMM requires two point-wise operations: PWM and PWA. In Kyber, a 256-term polynomial $a(x) = (a_0, a_1, \ldots, a_{254}, a_{255})$ is performed NTT process using two 128-point NTT, one for the even part $(a_{2i}(x) = (a_0, a_2, \ldots, a_{254}))$ and one for the odd part $(a_{2i+1}(x) = (a_1, a_3, \ldots, a_{255}))$. Thus, the PWM on Kyber is multiplying polynomials of the form $\hat{a}_{2i} + \hat{a}_{2i+1} \cdot X$. In study [48], Xing et al. proposed using the Karatsuba method to reduce the number of point-wise multiplications required to calculate $\hat{h} = \hat{f} \circ \hat{g}$ from five to four, as follows:

$$\hat{h}_{2i} = \hat{f}_{2i} \cdot \hat{g}_{2i} + \hat{f}_{2i+1} \cdot \hat{g}_{2i+1} \cdot \zeta^{2br(i)+1} \tag{11}$$

$$\hat{h}_{2i+1} = (\hat{f}_{2i} + \hat{f}_{2i+1}) \cdot (\hat{g}_{2i} + \hat{g}_{2i+1}) - \hat{f}_{2i} \cdot \hat{g}_{2i} - \hat{f}_{2i+1} \cdot \hat{g}_{2i+1} \tag{12}$$

The previous section mentioned that the K-RED method can be customized to change the output value. To remove the post-processing of the INTT stage, the output values of the PWM calculation should be $128^{-1} \cdot \hat{h}_{2i}$ and $128^{-1} \cdot \hat{h}_{2i+1}$, which can be achieved by applying the property $-13 \cdot 2^8 \equiv 1 (mod \ 3329)$ characteristic. The PWM architecture for Kyber is detailed in Figure 5. Initially, the calculations $\hat{f}_{2i} \cdot \hat{g}_{2i}$, $\hat{f}_{2i+1} \cdot \hat{g}_{2i+1}$, and $(\hat{f}_{2i} + \hat{f}_{2i+1}) \cdot (\hat{g}_{2i} + \hat{g}_{2i+1})$ all use K-DSPs without K-RED unit, resulting in bit widths 20. The calculation of $2 \cdot \hat{h}_{2i+1}$ is performed using modular addition, modular subtraction, and bit shifting. Subsequently, the K-RED unit is utilized to get the $128^{-1} \cdot \hat{h}_{2i+1}$ result. For the $2 \cdot \hat{h}_{2i}$ calculation, the PWM $\hat{f}_{2i+1} \cdot \hat{g}_{2i+1}$ is utilized with K-RED to provide the result in the ring $\mathbf{R}_{3329}$. The accumulated factor $k$ is eliminated from the PWM with pre-computed factor $k^{-1} \cdot \zeta^{2br(i)+1}$. Finally, K-RED is used again to determine the $128^{-1} \cdot \hat{h}_{2i+1}$ result.

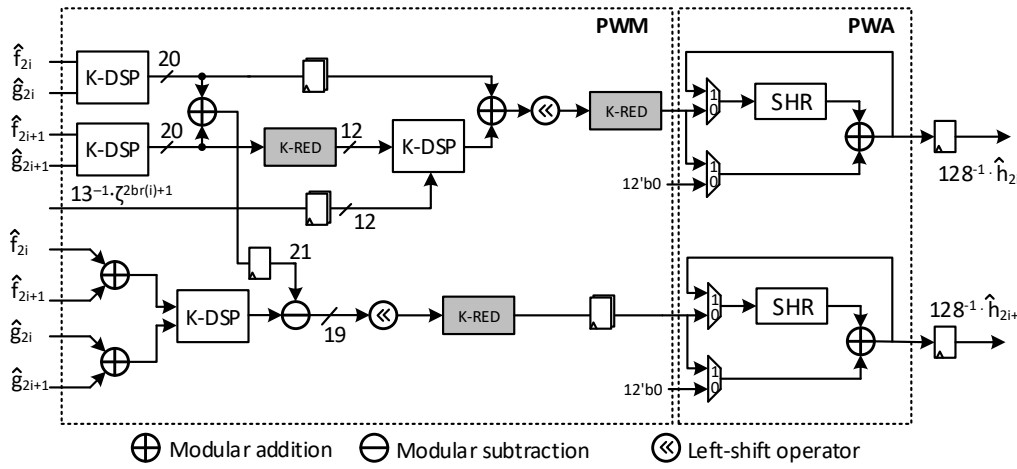

**Figure 5.** The proposed architecture of PWMM unit in Kyber implementation.

For PWA calculation, the architecture in Figure 5 shows an efficient and simple way to perform PWA using a shift register SHR with feedback and modular addition. During the initial calculation stage, the SHR takes in the output of PWM $\hat{a}_{00} \circ \hat{r}_0$ and adds value $12'b0$. In the following step, the SHR takes in the result of the previous addition and performs another addition with the outcome of the PWM $\hat{a}_{10} \circ \hat{r}_1$. This architecture requires no hardware cost changes when implementing different security levels of Kyber, $k = 2, 3,$ and $4$, respectively.

### 4.4. L-DSPs

NTT-based polynomial multiplication is a highly utilized and efficient method for implementing LBC systems, offering quasi-linear complexity of $O(N \cdot logN)$. To quickly calculate NTT using NWC, a prime number $q$ is carefully chosen to meet the condition $q \equiv 1 \ (mod \ 2N)$. This guarantees the existence of the $N$-th and $2N$-th primitive roots of unity (denoted as $\omega$ and $\zeta$) in the ring $\mathbf{R}_q$. Consequently, the parameters for the LBC algorithms require a relatively large value for $N$ and a relatively small modulus $q$ in form $q = k \cdot 2^m + 1$, where $2N \mid 2^m$ and $k \geq 3$ is considered small integer.

This study focuses on analyzing and proposing the DSP for commonly used parameter sets $(N, q)$ of LBC algorithms, as in Table 2 (referred to as L-DSP). The Kyber team has adopted the parameter pair $(256, 3329)$ since round 2 of the NIST competition. It is chosen to address increased bandwidth requirements resulting from removing public key compression. The advantage of using $q = 3329$ is that NTT-based polynomial multiplication can be performed quickly, leading to smaller noise. One drawback to this parameter set is that PWM calculation is not directly possible.

**Table 2.** Parameter sets $(N, q)$ of LBC algorithms.

| Typical Algorithm | N | q |
|---|---|---|
| CRYSTAL-Kyber (v2,v3) | 256 | 3329 |
| CRYSTAL-Kyber (v1) | 256 | 7681 |
| FALCON | 512/1024 | 12,289 |

Alternatively, $(256, 7681)$ is the smallest parameter pair that fully supports fast NTT computation while ensuring high security and the ability to perform direct PWM calculations. Due to this advantage, many current hardware implementations have adopted the prime $q = 7681$ to optimize their systems [11,46,49]. The remaining set (512/1024, 12,289) is utilized in the latest version of FALCON, where the polynomial degree $N$ can vary depending on the desired security level of the system.

The primary operations that use DSP for modular multiplication are butterfly and PWM. In the BFU architecture, a coefficient is multiplied by a pre-computed twiddle factor $(\omega)$. The accumulated $k$ can be merged to $\omega$ as $k^{-1} \cdot \omega$. The value of $N$ does not impact the K-RED architecture in L-DSP in the case of FALCON. Algorithm 3 outlines the steps for implementing L-DSP for butterfly computation in a comprehensive and detailed manner.

---

**Algorithm 3** L-DSP for Butterfly Unit

---

**Input:** n-bit integers: b, $\omega' = k^{-1} \cdot \omega$, prime $q$, and small integers $k, m$.
**Output:** $r \equiv b \cdot \omega \ (mod \ q)$
    **Stage 1: Karatsuba, Vedic multiplier**
 1: $t_0 = b_H \cdot \omega'_H$;
 2: $t_1 = b_L \cdot \omega'_L$;
 3: $t_2 = (b_H + b_L) \cdot (\omega'_H + \omega'_L)$;
    **Stage 2: Calculate the product**
 4: $p_0 = (t_0[n-1, \ldots, n+2-m])_{LUT}$;
 5: $p_1 = \{t_0, t_1\}$;
 6: $p_2 = t_2 - t_1 - t_0$;
 7: $p = p_0 + p_1 + p_2$;
    **Stage 2: K-RED Reduction**
 8: $d_0 = p[m-1, \ldots, 0]$;
 9: $d_1 = p[n+m-1, \ldots, m]$;
10: Return $(k \cdot d_0 - d_1)$

---

In the PWM operation, one crucial step involves multiplying the output by the value $N^{-1}$ to eliminate post-processing in the INTT. Notably, as the polynomial degree $N$ changes, the value of $N$ consistently satisfies the condition $2N \mid 2^m$, which ensures that the output can be computed directly as,

$$r \equiv z \cdot c \ (mod \ q) \equiv (-k \cdot 2^m) \cdot N^{-1} \cdot c \ (mod \ q) \tag{13}$$

with $z = 2^m / N$ and $-k \cdot 2^m \equiv 1 \ (mod \ q)$. The Equation (13) is equivalent to $r \equiv N^{-1} \cdot c \ (mod \ q)$. Additionally, since $z$ is a power of two, it can be easily multiplied by bit-shifting the input of the K-RED part. Algorithm 4 provides a detailed outline of the L-DSP implementation for PWM operation.

For multiplying coefficients with larger bit-width, it is possible to create an efficient multiplier design using 3-bit and 4-bit Vedic multiplier circuits as the basic building blocks.

---

**Algorithm 4** L-DSP for PWM Unit

---

**Input:** n-bit integers: a, b, prime q, and small integers $k, m, z = 2^i$, degree $N$.
**Output:** $r \equiv N^{-1} \cdot a \cdot b \ (mod \ q)$

    **Stage 1: Karatsuba, Vedic multiplier**
1: $t_0 = a_H \cdot b_H$;
2: $t_1 = a_L \cdot b_L$;
3: $t_2 = (a_H + a_L) \cdot (b_H + b_L)$;
    **Stage 2: Calculate the product**
4: $p_0 = (t_0[n-1, .., n+2-m-i])_{LUT}$;
5: $p_1 = \{t_0, t_1\}$;
6: $p_2 = t_2 - t_1 - t_0$;
7: $p = (p_0 + p_1 + p_2) \ll i$;
    **Stage 2: K-RED Reduction**
8: $d_0 = p[m-1, \ldots, 0]$;
9: $d_1 = p[n+m-1, \ldots, m]$;
10: Return $(d_1 - k \cdot d_0)$

---

## 5. Implementation Results

This study introduces and applies the proposed modular multiplication L-DSPs in BFU and PWMM units of the NTT-based accelerator in the LBC cryptosystem. These architectures are synthesized and place-and-routed using the Xilinx Vivado 2021.2 suite. The widely used Xilinx Artix-7 FPGA platform (part number XC7A100tfgg676-3) is selected to ensure a fair comparison with state-of-the-art hardware implementations. We introduce the hardware efficiency ($Eff.$) for a comparative analysis with previous works. A higher $Eff.$ value is desirable and can be calculated as follows,

$$Eff = \frac{Frequency \times No. \ of \ bits}{No. \ of \ LUTs} \ (Kbps/L) \tag{14}$$

The study [47] showed a normal DSP with an equivalent conversion rate of 100 SLICEs or 400 LUTs. We use this ratio for area comparison with other studies.

Table 3 shows the proposed K-DSP architecture for modular multiplication on CRYSTAL-Kyber, showcasing its speed and area. K-DSP can perform integer multiplication and modular reduction and operates at a frequency of 283 MHz, occupying an area of only 77 SLICEs, equivalent to 77% of a DPS. It is worth noting that all other studies listed in the comparison table use DSP for coefficient multiplication. The implementation results of the K-RED method are better than other methods when performing modular reduction. Notably, in study [36], by combining the K-RED and LUT methods, the operating frequency reached 300 MHz with an equivalent area of 50 (+400) LUTs. In [46], heavy multiplications are efficiently replaced with compact bit-wise operations and additions/subtractions based on an optimized Barrett algorithm. The hardware results achieved an operating frequency of 265 MHz and occupied an equivalent area of 81 (+400) LUTs. Study [33] utilizes a bit-reduce method that is complex and hardware costly. These results demonstrate that the proposed K-DSP architecture has further optimized modular multiplication, with significantly improved $Eff$ performance metrics of 1.86×, 2.25×, 2.46×, 2.57×, 4.1×, and 4.2× compared to studies [18,33,36,46,48,50], respectively.

**Table 3.** Comparison of the proposed modular multiplication DSP for Kyber with other approaches.

|  | Method | Freq (MHz) | LUTs | FFs | SLICEs | DSP | Eff. (Kbps/L) |
|---|---|---|---|---|---|---|---|
| **Proposed DSP** | K-DSP | 283 | 228 | 174 | 77 | 0 | 14,895 |
| [18] | K-RED | 222 | 59 (+400) | 33 | 19 (+100) | 1 | 5804 |
| [19] | K-RED | N/A | 54 (+400) | 30 | 18 (+100) | 1 | N/A |
| [33] | Bit-reduce | 159 | 142 (+400) | 79 | 0 (+100) | 1 | 3520 |
| [36] | K-RED, LUT | 300 | 50 (+400) | 34 | 15 (+100) | 1 | 8000 |
| [46] | Barret | 265 | 81 (+400) | 112 | 0 (+100) | 1 | 6611 |
| [48] | Barret | 161 | 135 (+400) | 96 | 0 (+100) | 1 | 3611 |
| [50] | K-RED | 232 | 59 (+400) | 70 | 24 (+100) | 1 | 6065 |

The results of the K-DSP-based BU implementation for Kyber are displayed in Table 4. The proposed BU architecture comprises one K-DSP, an adder, and a subtractor for CT and GS butterfly operations. The operating frequency of this architecture reaches 283 MHz and takes an area of 104 SLICEs, slightly equal to one DSP. The conventional unified BU architecture uses two DSPs to perform distinct multiplications, adders, and subtractors for CT and GS calculations. The studies [33,48] used this architecture and recorded low operation frequency and high hardware resources of 159 MHz for 774 (+800) LUTs and 161 MHz for 647 (+800) LUTs, respectively. Other studies have built a reduced architecture using only one DSP for multiplication. With the downsized BU architecture, the study [46] has an operating frequency of 265 MHz and consumes 186 (+400) LUTs. It is important to mention that multiplication can impact speed and area. In study [51], a standard implementation of Kyber's reference code, modular multiplication consumed more DSPs due to applying both the Barret and Montgomery algorithms. The proposed BU architecture based on K-DSP significantly improves the $Eff.$ index compared to the research studies [18,27,33,46,48,50]. The improvement is 2.06×, 2.52×, 3.01×, 3.26×, 8.39×, and 9.22× times, respectively.

**Table 4.** Implementation results of the proposed BU for CRYSTAL-Kyber and comparison with previous studies.

|  | Freq (MHz) | LUTs | FFs | SLICEs | DSPs | Eff. (Kbps/L) |
|---|---|---|---|---|---|---|
| **This work** | 283 | 304 | 234 | 104 | 0 | 931 |
| [18] | 222 | 200 (+400) | 179 | 0 (+100) | 1 | 370 |
| [27] | 229 | 440 (+400) | 499 | 0 (+100) | 1 | 286 |
| [33] | 159 | 774 (+800) | 394 | 317 (+200) | 2 | 101 |
| [46] | 265 | 186 (+400) | 172 | 0 (+100) | 1 | 452 |
| [48] | 161 | 647 (+800) | 501 | 0 (+200) | 2 | 111 |
| [50] | 208 | 274 (+400) | 181 | 0 (+100) | 1 | 309 |
| [51] | N/A | 177 (+2000) | 0 | 0 (+500) | 5 | N/A |

Table 5 shows the effectiveness of the DSP design method for core operations modular multiplication and butterfly in the NTT accelerator of the LBC cryptosystem. We have developed L-DSPs and BUs architectures for prime $q$ values in $q = k \cdot 2^m + 1$, specifically for 3329, 7681, and 12,289. When using L-DSPs, the operating frequencies for $q$ = 3329, 7681,

and 12,289 are 283 MHz, 272 MHz, and 256 MHz, respectively. The hardware resources needed for L-DSPs are less than or equal to one DSP, which results in a percentage of the area used of 77%, 87%, and 101% DSP, respectively. On the other hand, using BUs results in hardware resource improvements of 104%, 120%, and 136% DSP for the same $q$ values, respectively. The operating frequencies for BUs are 283 MHz, 260 MHz, and 250 MHz, respectively. All L-DSPs and BUs architectures have less than 400 LUTs hardware resources, equivalent to one DSP.

**Table 5.** Implementation results of the proposed L-DSPs and BU for typical cases of prime $q$ in LBC cryptosystem.

| Architecture | Prime Number $q$ | Freq (MHz) | LUTs | FFs | SLICEs |
|---|---|---|---|---|---|
| L-DSP | 3329 | 283 | 228 | 174 | 77 |
| | 7681 | 272 | 277 | 188 | 87 |
| | 12,289 | 256 | 306 | 262 | 101 |
| BU | 3329 | 283 | 304 | 234 | 104 |
| | 7681 | 260 | 362 | 253 | 120 |
| | 12,289 | 250 | 393 | 332 | 136 |

The results of implementing the PWMM architecture in Kyber are presented in Table 6. Other studies utilize BRAM [46] or FIFO [47] to temporarily store the accumulation results, leading to further hardware resource consumption for the PWA operation. In order to optimize the NTT accelerator pipelines, our architecture is designed to handle both PWM and PWA operations. Two different architectures, namely one-PWMM and two-PWMM, are designed and achieve the highest operating frequency of 275 MHz with cycles and an area of 128 Clks/1123 LUTs and 64 Clks/2297 LUTs. To be more precise, the hardware resources take fewer SLICEs than the conversion value of 4 and 8 DSPs, which are equivalent to 386 and 797 SLICEs, respectively. In studies [34–36], the PWM execution is performed using a shared hardware architecture with BUs. The highest operating frequencies are achieved in studies [35,36], reaching 300 MHz and 303 MHz, respectively. This is primarily due to the use of efficient modular reduction modules. In study [46], an architecture for two-PWM calculation was proposed on Kyber, using 8 DSPs for the multiplications. The operating frequency of this architecture reaches 265 MHz and consumes hardware resources of 749 (+3200) LUTs respectively. Our architecture significantly reduces hardware resources and improves efficiency. The proposed PWMM design reduces ATP(area time product) by 33.8%, 47.8%, 67.3%, and 71.2% compared to [34–36,46], respectively.

**Table 6.** Implementation results of the proposed PWMM unit for CRYSTAL-Kyber.

| | Freq (MHz) | LUTs | FFs | SLICEs | DSPs | Cycles | ATP | Modes |
|---|---|---|---|---|---|---|---|---|
| This work [1] | 275 | 1123 | 1061 | 386 | 0 | 128 | 180 | PWM, PWA |
| This work [2] | 275 | 2297 | 2081 | 797 | 0 | 64 | 185 | PWM, PWA |
| [34] | 200 | 1740 (+1600) | 643 | 575 (+400) | 4 | 128 | 624 | PWM |
| [35] | 300 | 1154 (+800) | 1031 | 445 (+200) | 2 | 256 | 550 | PWM |
| [36] | 303 | 1170 (+1600) | 1164 | 416 (+400) | 4 | 128 | 345 | PWM |
| [46] | 265 | 749 (+3200) | 1103 | 325 [3] (+800) | 8 | 64 | 272 | PWM |

Area-Time-Product (ATP) = ((Cycles $\cdot 10^6$) $\times$ SLICEs) / Freq. [1,2] Number of PWMM units. [3] Since a single slice contains four LUTs and eight FFs, we convert SLICEs = $\frac{1}{4} \times$ LUTs + $\frac{1}{8} \times$ FFs.

## 6. Conclusions

In this paper, we present a method for designing compact and efficient specialized hardware implementations for modular multiplication in LBC systems. Using the proposed approach, we have designed and implemented core architectures within NTT accelerators for polynomial multiplication. The optimization of the BU architecture is performed to completely eliminate the need for post-processing in INTT. Consequently, the BU architecture, when simultaneously executing NTT and INTT, has a footprint equivalent to that of a conventional BU architecture. Additionally, we propose a hardware design for implementing PWMM for the Kyber algorithm. The proposed architecture can perform both PWM and PWA calculations for all security levels without requiring additional temporary memory, such as RAM or FIFO buffers.

Furthermore, we have demonstrated the effectiveness of the proposed method by designing it for common prime parameters $q$, used in various LBC algorithms. FPGA-based implementation results show the outperforming hardware efficiency of the K-DSP, L-DSP, BU, and PWMM architectures compared to existing implementations. The findings of this paper can further optimize NTT accelerators with pipeline or iterative configurations. Therefore, the proposed architectures represent an important step toward designing compact and high-performance post-quantum lattice-based cryptography systems on hardware platforms.

**Author Contributions:** Supervision, C.-K.P. and T.-T.H.; methodology, T.-H.N.; investigation, T.-H.N.; writing—original draft preparation, T.-H.N.; writing—review and editing, T.-H.N. All authors have read and agreed to the published version of the manuscript.

**Funding:** This research received no external funding.

**Institutional Review Board Statement:** Not applicable.

**Data Availability Statement:** Not applicable.

**Acknowledgments:** This work was supported by the University of Electro-Communications (UEC) via the Support for External Funds Acquisition by Early Career Scientists program.

**Conflicts of Interest:** The authors declare no conflict of interest.

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
