# Peer review of "A High-Efficiency Modular Multiplication Digital Signal Processing for Lattice-Based Post-Quantum Cryptography"

_cryptography, doi:10.3390/cryptography7040046_

Round 1

Reviewer 1 Report

The paper introduces a DPS specifically designed for CRYSTALS-Kyber algorithm. It merges different tecniques (Karatsuba, Vedic and K-RED method) to optimize the modular multiplication and reduce the area occupation of the multiplier. In addition, the paper proposes an implementation of the Point Wise Modular Multiplication (PWMM) of Kyber that obtain good results in terms of area consumption. 

The techniques proposed in the paper are original and relevant for the scientific community.

Here some suggestions to improve the quality of the paper: 

1.      Figure 1 shows the architecture of the modular multiplier based on Karatsuba algorithm. Since the inputs of the multiplication are 12-bit values, the authors should better specify why the output of the Karatsuba stage is 20-bit (instead of 24-bit). In general, the bit width described in Figure 1 should be better justified.

2.      Improve some sentences such as:

-line 13: Especially the Point-Wise Matrix Multiplication unit 13 (PWMM) of CRYSTAL-Kyber is proposed only consumes the area of 386 SLICEs.

 -line 26: Most are lattice based etc...

3.      It could be interesting to compare the performance of the proposed UBU with the one of this paper "A RISC-V Post Quantum Cryptography Instruction Set Extension for Number Theoretic Transform to Speed-Up CRYSTALS Algorithms," in IEEE Access, since this is a standard implementation that follows the reference code of Kyber; the proposed work applied different optimization strategies and the achieved results can be compared to show the advantages of the proposed implementation.

4.     The authors can also compare their results with this work: M. Li, J. Tian, X. Hu and Z. Wang, "Reconfigurable and High-Efficiency Polynomial Multiplication Accelerator for CRYSTALS-Kyber," in IEEE Transactions on Computer-Aided Design of Integrated Circuits and Systems, since this is probably the newest article related to hardware acceleration of Kyber algorithm.

Reviewer 2 Report

The paper is interesting and the content may be useful for the community. I think it could be published in Cryptography. The presentation of the manuscript could be further improved. For example, The text between Eq. (1) and (2), (4) and (5), (5) and (6) should not be a new paragraph.  

Reviewer 3 Report

The paper presents rather good results, but the research is not aligned to the state-of-the-art. 

Please check newer papers regarding Barrett and Montgomery type algorithms. At least in the case of Barrett's algorithm, relevant speed-ups have been proposed during the last 10 years (including last year). As a natural follow up, please compare your results with up to date papers and report related work accordingly.

References need to be updated, there are many newer articles to cite (see the above also).

English is fine in general, but a grammar checker could be used for better results.

Reviewer 4 Report

Good paper. Original, novel and clearly presented. Some suggestions for the final version:

1) I've counted more than 20 acronyms + some wrongly written acronyms. I would suggest to add an explicit acronym list in the paper.

2) BFU acronym in the paper should be wrong.

3) IND-CCA acronym not defined.

4) Definition of acronym CBD probably unnecessary. Used only once.

5) Line 155: could you please formaly define the "bit reversal" operation?

6) Equations 2 and 3: some missing parentheses.

7) Line 220: I guess should be "L-DSP"?

8) Figure 1: Could you add sizes of input lines?

9) Equation 11. Circle is the point-wise multiplication for polynomials. But in this equation in matrix format, the multiplication is matrix multiplication. I guess the standard matrix product symbol should be used.

10) Equation 15. Units of the equation should be (Kbps/L) instead of (Kbps/CL). Kbps means number of bits produced per second so it embeds the number of cycles required by the processor.

Round 2

Reviewer 3 Report

The authors have addressed my comments, clarifying the points I raised.